# Stress-Induced Immunomodulation in Low and High Reactive Sheep

**DOI:** 10.3390/ani9030104

**Published:** 2019-03-20

**Authors:** Mhairi Sutherland, Suzanne Dowling, Richard Shaw, Jackie Hickey, Diane Fraser, Catherine Cameron, Ian Sutherland

**Affiliations:** 1AgResearch Ltd., Ruakura Research Centre, Hamilton 3240, New Zealand; Suzanne.Dowling@agresearch.co.nz (S.D.); richard.shaw@agresearch.co.nz (R.S.); Catherine.Cameron2@agresearch.co.nz (C.C.); Ian.Sutherland@agresearch.co.nz (I.S.); 2Environmental and Animal Sciences Practice Pathway, Unitec Institute of Technology, Private Bag 92025, Auckland 1142, New Zealand; petsclare@gmail.com (J.H.); dfraser@unitec.ac.nz (D.F.)

**Keywords:** CarLA IgA, Ewe, heart rate, isolation, temperament, welfare

## Abstract

**Simple Summary:**

Proper functioning of the immune system is fundamental to maintain animal health; however, several factors can modulate an animal’s immune system, including stress. Farm animals can experience multiple stressors throughout their lifetime, therefore there is a need to know how stress can impact their immune response. Moreover, temperament can affect how an animal responds to a stressor, both behaviorally and physiologically. CarLA is a protective antigen against gastrointestinal nematodes; we wanted to evaluate if there was a relationship between stress and temperament on the CarLA response in ewes. We found that both a 0.5 h and 23 h stressor appeared to have an immunosuppressive effect on CarLA IgA but not total IgA concentrations in ewes and there was some indication that CarLA concentrations were also affected by ewe temperament. More research is needed to understand what these immunosuppressive effects of stress on CarLA IgA concentrations mean in relation to sheep immunity to parasitism using farm relevant stressors.

**Abstract:**

The objective of this study was to investigate the relationship between stress and temperament on the humoral immune response of ewes. Eighty ewes were allocated to one of four treatment groups in a 2 × 2 factorial design (*n* = 20 ewes/treatment): low (LR) and high (HR) reactive ewes were either exposed to no stress (CON) or were visually isolated (STRESS). Ewes remained in treatment pens for 23 h: heart rate was measured continuously, and saliva samples were collected prior to testing and at 0.5 h and 23 h for measurement of cortisol, CarLA IgA and total IgA concentrations. After the first 0.5 h, heart rate was elevated, and cortisol concentrations tended to be higher, whereas CarLa IgA concentrations were lower in STRESS than CON ewes. Similarly, after 23 h, cortisol concentrations remained elevated and CarLA IgA concentrations remained lower in STRESS than CON ewes. Interestingly, total IgA concentrations were not influenced by a 0.5 h or 23 h stressor. Overall, CarLA IgA concentrations were lower in HR than LR ewes at 0.5 h, but there was no significant stress × temperament interaction. Therefore, stress appears to have an immunosuppressive effect on CarLA IgA but not total IgA concentrations in ewes.

## 1. Introduction

The proper functioning of the immune system is fundamental to maintaining animal health; however, several factors can modulate an animal’s immune system, either positively or negatively, including stress [1]. Unfortunately, stress can be difficult to define. Moberg [2] described stress on the basis of duration and frequency of the stressor; thus, acute stress was defined as a brief experience that involves a single stressor, while chronic stress comprised of exposure to a series of acute stressors. Tewes [3], meanwhile, stated that stressors may be interpreted by the quality, intensity, duration and frequency of the stressor, but concluded that an individual animal’s reaction determines ‘stress’. Moreover, evidence suggests that an acute stressor can enhance the immune system while a chronic stressor can result in immunosuppression in human and animal studies [4,5]. Sheep can potentially experience multiple stressors throughout their lifetime, both acute and chronic, such as handling, heat or cold stress, under-nutrition and transport. Therefore, it would be of interest to investigate whether stress affects the immune response of sheep and, in addition, whether duration of the stressor impacts this response.

Immunoglobulin (Ig) A is one of the major antibodies associated with the humoral mucosal immune system in mammals and birds [6]. It can bind selectively to antigens such as viruses and toxins and can act to prevent organisms from penetrating or interacting with the mucosal epithelium [6,7]. Secretory IgA (S-IgA) has been used as an indicator of mucosal immunity in animal and human studies; furthermore, it has been shown to be sensitive to stress. Stress can have an enhancing or suppressive effect on S-IgA concentrations depending on stressor type, duration and timing of sampling (reviewed in Staley et al. [6]). Because of this relationship between stress and S-IgA, it has been suggested that S-IgA could be used as a physiological biomarker in studies designed to investigate the relationship between stress and immunity [6].

Parasitism with gastrointestinal nematodes (GIN) is a common health issue in sheep and can result in reduced productivity [8]. Gastrointestinal nematodes in sheep are commonly controlled by administering broad-spectrum anthelmintic drugs [8]. However, alternative strategies to control for parasitism with GIN that are less reliant on anthelmintic drugs are being investigated including breeding sheep that are naturally resistant to GIN or the development of anti-parasite vaccines [9]. Research has shown that sheep challenged with GIN produce an IgA response against a carbohydrate larval surface antigen (CarLA) which occurs on trichostrongylid nematode species [10,11]. Moreover, studies have demonstrated that CarLA is a protective antigen against GIN [11,12,13] and sheep identified as having high levels of anti-CarLA IgA had lower fecal egg counts than animals with low or undetectable levels [9,13]. Therefore, it would be of interest to evaluate if anti-CarLA IgA is similarly affected by stress as total S-IgA, and whether CarLA IgA could also be used as a biomarker to evaluate the relationship between stress and immunity in sheep.

In addition, how an animal perceives or responds to a stressor could potentially impact its ability to cope with an immune challenge (e.g., by infectious organisms). There are many factors that can influence how an individual perceives or responds to a stressor, such as previous experience, sex, genetics and temperament. Temperament is a term that can be used to describe individual differences in how animals respond to an environmental change or challenges [14] and animals are usually divided into sub-groups dependent on their behavioral response to a challenge, e.g., calm or temperamental [15], more or less active [16] and high or low avoidance [17]. Sheep categorized as being ‘more’ active crossed more zones, were less fearful and had lower cortisol concentrations in response to the presence of a human in an arena test than ‘less’ active sheep [16]. Moreover, dairy cattle categorized as being ‘high’ responders, based on their responsiveness to humans, had higher heart rates and lower milk yields when milked in a novel milking parlor compared with ‘low’ responder cows [18]. Therefore, if temperament influences stress responsiveness, then there could potentially be a relationship between animal temperament and immunocompetence. Establishing a link between animal temperament, stress responsiveness and immunity could have significant implications in animal production systems.

Biomarkers for stress and immunity are commonly measured using blood collection techniques. However, some of these biomarkers can now be measured in saliva, which could potentially provide a less invasive method of measurement. Cortisol is commonly used as a biomarker for stress in animals and has been successfully measured in the saliva of sheep in response to several different stressors [19,20,21]. Measurement of anti-CarLA IgA is possible in saliva and provides a practical, rapid and relatively non-invasive method to measure natural immunity to GIN in sheep [9]. Furthermore, Staley et al. [6] suggested S-IgA as a non-invasive biomarker for evaluating the interplay between stress and the immune system, particularly because S-IgA is secreted at all mucosal surfaces and therefore reflects the functional status of the mucosal immune system. In addition, S-IgA concentrations can be influenced by both physical and psychosocial stress [6]. Therefore, the objective of this study was to investigate the relationship between stress and temperament on a selected, specific humoral immune response of ewes after exposure to a short (0.5 h) and longer term (23 h) stressor.

## 2. Materials and Methods

### 2.1. Animals

Eighty Coopworth ewes (56.7 ± 3.38 kg body weight; mean ± SD), approximately 1 year of age, were selected and transported from the Whatawhata Research Station to the AgResearch Ruakura Research farm in the Waikato region, New Zealand. The study was conducted between March and April 2011 and was approved by the Ruakura Animal Ethics Committee (Protocol No. 12550) prior to the commencement of the study.

### 2.2. Open-Field Test

An open-field (OF) test was used to assess the temperament of all 80 ewes prior to the visual isolation test using methodology adapted from Doyle et al. [22]. The OF test was performed between 10 days and 12 days prior to the visual isolation test. On the test day, heart rate monitors were attached to the ewes (as described below) 30 min prior to entering the OF test so that five min of baseline heart rate data could be obtained prior to testing. Ewes were then moved individually into the OF arena. The OF arena measured 3 m × 3 m and had 1 m × 1 m squares (zones) marked and numbered one to nine on the concrete floor. Two sides of the arena had black fabric attached to the walls to prevent visual contact with other sheep in the adjoining holding pens. The other two sides of the arena (one solid wood and one metal gate) did not allow visual contact with other sheep. Each ewe was left in the arena for 5 min. An observer was seated above the arena and situated behind a wooden wall located 3 m from the arena so that the behavior of the sheep could be clearly observed without the sheep seeing the observer. Ewe behavior was scored continuously during the 5 min observation period in the OF arena and the number of squares entered, the number of vocalizations, escape attempts, urination and defecation events were recorded (behaviors are described in Table 1). At the end of the 5 min test period, the ewe was moved out of the arena, the arena was hosed down, and the next ewe was moved into the arena for testing.

Continuous heart rate (beats/min) was recorded 30 min prior to the ewe entering the OF arena and during the 5 min OF test period using Polar heart rate monitors (S810i™, Polar Electro Oy, Helsinki, Finland). On the day of testing, ewes were restrained manually and the wool around the girth was clipped and the heart rate monitor attached. After recording was completed, heart rate data were downloaded to a computer for analysis. Baseline heart rates were analyzed for 5 min while the ewe was standing in the holding yards prior to entering the OF arena, and for the 5 min period they were being tested in the arena. Before analysis, a correction function within the Polar software (Polar Precision Performance Software, version 4.03), set on default parameters, was used to correct for any artefacts, and only data with an error rate of less than 5% were included in the analysis.

After all ewes were tested in the OF arena, the frequency of all the behaviors performed by each individual ewe were summed to give each animal an overall reactivity score (Table 1). The reactivity score in combination with the heart rate response to the OF test was then used to categorize ewes as either low (LR) or high (HR) responders using similar temperament categories as described in Beausoleil et al. [16]. Each ewe was ranked based on their reactivity score and heart rate response: ewes that performed fewer vocalizations and escape attempts, entered less squares and had a lower heart rate response to the OF test were categorized as LR. Conversely, ewes with high reactivity scores and heart rate response were categorized as HR. The 40 ewes with the lowest reactivity scores and heart rate response were classified as LR and the 40 ewes with the highest scores and heart rate response as HR.

### 2.3. Visual Isolation Test

After temperament testing, sheep were allocated to one of four treatment groups in a 2 × 2 factorial design (*n* = 20 ewes/treatment): LR and HR ewes were either exposed to no stress (CON) or were visually isolated (STRESS). Ewes remained in the treatment pens for 23 h. The study was conducted over five consecutive days; eight ewes per treatment were tested on each treatment day.

Ewes subjected to visual isolation were housed in individual pens (1.3 m × 1 m) located in an enclosed research facility. The walls of the pens were covered in black fabric to prevent visual contact between ewes housed in adjoining pens and the walls were high enough so that the ewes could not jump out. The pens were raised 0.5 m off the ground and had metal mesh flooring covered with perforated rubber mats to allow for easy drainage of excrement.

Control sheep were housed in groups of eight in pens measuring 3.6 m × 3.6 m in the same research facility as the STRESS ewes. Pens were raised 1 m off the ground and the flooring consisted of metal mesh which allowed for easy drainage of excrement. Both STRESS and CON sheep had access to water ad libitum.

On the day of testing, colored collars were placed around the neck of each ewe for easy identification of individuals. Heart rate monitors were attached (as described above) to a subset of ewes (*n* = 10/treatment; *n* = 2/treatment/test day) 30 min prior to entering the test pens. Heart rate was measured on only a subset of animals due to logistical reasons. A saliva sample was collected from all ewes 5 min prior (baseline) to them entering the treatment pens, and then again 0.5 h (short term stressor) and 23 h (long term stressor) after testing commenced. To collect saliva samples, ewes were restrained and a cotton dental swab (100% Absorbent Cotton Size No. 2. Alan & Co., Verviers, Belgium) clasped in long-nose surgical forceps was placed between the cheek and gums at the side of the mouth. The swab was then gently manipulated for approximately 10 s. The collected swab was then placed into a 5 mL vial (LabServ Manufactured Plastics, Auckland, New Zealand) and kept cool until frozen. Saliva was extracted from the cotton swabs by centrifugation for 5 min at 2000 × *g*. All saliva samples were analyzed for CarLA IgA and total IgA concentrations as described below. Only 40 saliva samples (*n* = 10/treatment) were analyzed for total cortisol concentrations due to logistical reasons.

#### 2.3.1. Behavior

During the first 0.5 h that animals were in the test pens, ewe behavior was scored continuously from live observations by four trained technicians. The number of vocalizations, escape attempts, urination and defecation events were recorded. Due to the low occurrence of behaviors, the frequency of all behaviors was summed to give an overall behavioral score for each animal. Description of behaviors are presented in Table 1.

#### 2.3.2. CarLA IgA

Saliva samples were assayed to measure specific anti-CarLA IgA antibody concentrations as described in Shaw et al. [9,13]. Briefly, extracted saliva was diluted 1/20 in sample dilution buffer and stored at −20 °C until assayed. To assay for specific anti-CarLA IgA antibodies, the immunoassay was incubated overnight at 4 °C with 100 µL/well of purified CarLA in phosphate buffered saline (PBS). The plates were washed five times with reverse osmosis purified water containing 0.1% (w/v) Tween 20 (W-T20) then blocked for 30 min at room temperature with 5% skim milk powder in 10 mM phosphate buffer, 0.65 M saline, pH 7.2 containing 0.5% Tween 20. Plates were washed five times with W-T20. Diluted saliva was added to ELISA plate wells in duplicate then incubated for 2 h at 37 °C. Plates were then washed six times with W-T20. Rabbit anti-sheep IgA conjugated with horseradish peroxidase (Bethyl Laboratories Inc., Montgomery, TX, USA), diluted 1/4000 with ELISA buffer, was added to each well and incubated for 2 h at 37 °C. Plates were then washed seven times as above, and 100 µL/well of freshly prepared substrate added. The reaction was allowed to develop for 20 min at room temperature, and then stopped with 50 µL/well of 2 N sulfuric acid. The absorbance was measured at a wavelength of 450 nm. The concentration of specific anti-CarLA IgA was expressed as units/mL [13]. Within- and between-assay coefficients of variation were 14.3% and 18.7%, respectively.

#### 2.3.3. Total IgA

Saliva samples were collected and prepared as above. Immunoassay plates (Nunc Maxisorp 43041) were incubated overnight at 4 °C with 100 µL/well of anti-sheep IgA (Bethyl A130-108A) at 1 µg/mL in PBS. Plates were washed six times with PBS plus 0.05% (w/v) Tween 20 (PBS-T20) then blocked for 60 min as above. Plates were washed five times with PBS-T20. Saliva samples at 1/20 were further diluted 1/25 to get a 1/500 dilution. Diluted saliva was added to ELISA plate wells in duplicate but at two dilutions, 50 µL plus 50 µL sample dilution buffer (1/1000) and 25 µL plus 75 µL sample dilution buffer (1/2000) then incubated for 2 h at 37 °C. Plates were then washed six times with PBS-T20 with 5 min between each second wash. Rabbit anti-sheep IgA conjugated with horseradish peroxidase (Bethyl Laboratories Inc., Montgomery, TX, USA), diluted 1/75,000 with ELISA buffer, was added to each well (100 µL) and incubated for 2 h at 37 °C. Plates were then washed six times with PBS-T20 as above. Plates were developed with tetramethyl benzidine substrate as for the CarLA assay for 15 min. The same standard and method as used for the CarLA IgA assay was used for determining total IgA concentrations [13]. The effective range of this standard curve was 20–1600 units/mL for samples diluted 1/1000. Internal standards as used in the CarLA IgA assay were diluted 1/50 and used on ELISA plates to monitor assay variation. The within-assay coefficients of variation was 18.0%.

#### 2.3.4. Cortisol

Saliva samples were analyzed for cortisol concentrations by a commercial laboratory using standard quality control methodologies. Cortisol concentrations were determined using a commercially available enzyme immunoassay kit (catalogue #1-3004, Salimetrics, State College, PA, USA). The sensitivity of the assay is <0.07 ng/mL.

### 2.4. Statistical Analysis

Data from the OF test are presented descriptively. The data from the visual isolation test were analyzed using Genstat 18th edition (VSN International LTD, 2015) as a linear mixed model with day of the test as a random effect, and temperament and stress treatments and their interaction as fixed effects. CarLA and IgA data were log-transformed and vocalization and behavioral data were square-root transformed in order to meet the normality assumptions of the analyses. Baseline values for CarLA, IgA and cortisol were used as covariates for analysis of the 0.5 h and 23 h post-treatment values to compensate for baseline variation. Back-transformed log means and standard errors are presented with *p*-values from the log-transformed analyses. For square-root transformed data, untransformed means and standard errors are presented with *p*-values from the square-root transformed analyses.

## 3. Results

### 3.1. Open-Field Test

High reactive ewes vocalized more, performed more escape attempts, entered more squares and had a greater increase in heart rate than low reactive ewes in response to the OF test (Table 2). However, low reactive ewes defecated more in the OF test than high reactive ewes (Table 2).

### 3.2. Visual Isolation Test

During the first 0.5 h of the visual isolation test day, STRESS ewes had higher behavioral scores (*p* < 0.001) and vocalized more (*p* < 0.001) than CON ewes. During the first 0.5 h, neither the frequency of vocalizations nor the behavioral score was affected by ewe temperament, and there was no stress × temperament interaction (stress × temperament interaction: *p* = 0.790 and *p* = 0.575 for the behavioral score and frequency of vocalizations respectively) (Table 3).

Heart rate was higher (*p* < 0.001) in STRESS than CON ewes during the first 0.5 h of the visual isolation test, but there was no difference (*p* = 0.823) between treatments during the last 30 min of the 23 h test period, irrespective of temperament (Figure 1). Heart rate was not affected by ewe temperament and there was no stress × temperament interaction at 0.5 h (*p* = 0.399) or 23 h (*p* = 0.145) (Table 3).

CarLA IgA concentrations were lower than baseline values after a 0.5 h and 23 h test period for both CON and STRESS ewes, however this decline was greater in STRESS ewes (% change from baseline: at 0.5 h (CON: 32.7%, STRESS: 69.1%); at 23 h (CON: 30.9%, STRESS: 52.9%)). CarLA IgA concentrations were lower in STRESS ewes at 0.5 (*p* < 0.001) and 23 h (*p* < 0.001) after exposure to a visual isolation test compared with ewes housed in control pens (Figure 2). There was no difference (*p* = 0.641) in baseline CarLA IgA concentrations between LR and HR ewes. CarLA IgA concentrations were lower in HR than LR ewes at 0.5 h (*p* = 0.022) but not 23 h (*p* = 0.147), irrespective of stress treatment (Figure 3). There was no stress × temperament interaction at 0.5 h (*p* = 0.135) or 23 h (*p* = 0.414) (Table 3).

Total IgA concentrations were not affected (*p* > 0.05) by stress or ewe temperament after a 0.5 h or 23 h test period (Table 3). There was no stress × temperament interaction at 0.5 h (*p* = 0.673) or 23 h (*p* = 0.729) (Table 3).

Cortisol concentrations tended to be higher in ewes at 0.5 h (*p* = 0.098) and were higher at 23 h (*p* = 0.031) after exposure to a visual isolation test, compared with ewes housed in control pens (Figure 4). Cortisol concentrations were not affected by ewe temperament and there was no stress × temperament interaction at either 0.5 h (*p* = 0.536) or 23 h (*p* = 0.495) (Table 3).

## 4. Discussion

In the present study, stress appeared to have an immunosuppressive effect on salivary CarLA IgA concentrations in ewes, but total salivary IgA concentrations were not affected. Furthermore, there appeared to be no relationship between ewe temperament and stress on either CarLA IgA or total IgA activity. Isolating sheep from their conspecifics in a novel environment can be considered a good test of fear [23] and has been shown to elicit a behavioral and physiological stress response [24,25,26]. Similarly, in the present study, sheep vocalized more, heart rate was elevated and cortisol concentrations tended to be higher in response to a 0.5 h visual isolation test; therefore, it appears as though both the sympathetic nervous system and the hypothalamic–pituitary–adrenal (HPA) axis were activated in response to this stressor. After 23 h in the visual isolation test, cortisol concentrations were still elevated in ewes, but not heart rate. This is to be expected, as activation of the sympathetic nervous response occurs quickly and often returns to baseline rapidly [2]. However, continued elevation of cortisol concentrations in STRESS ewes suggests that these ewes had not habituated to the visual isolation test and were still experiencing stress at this time. Originally, we postulated that an acute stressor may enhance the immune response while a chronic stressor could cause immunosuppression, but it is unlikely that a continuous 23 h stressor is sufficient to be categorized as chronic. In fact, Dhabhar and McEwen [27] defined an acute stressor as lasting for a period of minutes to hours, whereas a chronic stressor persists for several hours per day for weeks or months. Therefore, in the present study, both time points measured can only be considered as an acute stress response. Further studies investigating the impact of chronic stress on CarLA IgA concentrations are required.

The rapid change in salivary CarLA IgA concentration in response to a 0.5 h isolation stress in sheep in the present study was not surprising as stressor durations of less than 30 min have been shown to cause changes in S-IgA concentrations in animal [28,29] and human [30] studies. The short time frame in which an acute stressor can cause changes in S-IgA concentrations is likely due to changes in IgA secretion mediated by the autonomic nervous system [30,31]. However, contradictory to our findings, acute stressors have mostly been reported to cause an increase in S-IgA concentrations (reviewed by Staley et al. [6]). Some studies though have reported that acute stress can have either an enhancing or a suppressive effect on S-IgA levels depending on the type of stressor and the how the stressor stimulates the autonomic nervous system. Bosch et al. [30] found that an acute active coping stressor (a time-paced memory test) elicited an increase in heart rate and blood pressure, a decrease in vagal tone and induced an increase in S-IgA concentrations; however, an acute passive coping stressor (viewing a distressing video) produced a decrease in heart rate, enhanced vagal tone and resulted in a reduction in S-IgA concentrations. Similarly, Willemsen et al. [31] found that an acute active coping test (mental arithmetic test) resulted in an increase in S-IgA concentrations and an acute passive coping stressor (a cold pressor test) elicited a decrease in S-IgA levels. Therefore, the stressor used in the present study may have activated aspects of the autonomic nervous system similar to those activated by the passive coping stressors in the human studies, hence resulting in a decrease in salivary CarLA IgA concentrations. More research is needed to understand the relationship between acute stress and the mucosal immune response in sheep.

Contradictory to our results, total salivary IgA concentrations increased in response to a 20 min stressor in 6–8-week-old pigs [28] and to a 10 min stressor in 7-week old puppies [29]. Although similar to our results, both of these studies found that cortisol concentrations were elevated in response to the stressor. In the same study, Svobodová et al. [29] tested total salivary IgA and cortisol concentrations in adult dogs in response to a four minute stressor and found that cortisol increased whereas IgA concentrations decreased. In the present study, adult animals were also tested, therefore age, differences in the maturation of the immune response, or how animals perceive stress may account for the difference in the IgA response between studies. In addition, it appears as though activation of the HPA axis and secretion of cortisol may not be responsible for the changes in the IgA concentrations alone, as cortisol was elevated in response to stress in all studies. Both glucocorticoids and catecholamines have been shown to be involved in the reduction in intestinal IgA concentrations in response to a chronic restraint stress in mice [32]. However, it appears as though the autonomic nervous system may be more involved in the regulation of the mucosal immune response during exposure to acute stressors. More research is needed to understand the mechanisms involved in the immunosuppressive effect of stress on salivary CarLA IgA concentrations and what it means in regard to ewe immunity to parasitism and what effect farm relevant stressors (e.g., tail docking, under nutrition, heat stress) may have on CarLA IgA concentrations and immunity to parasites in sheep.

In the present study we measured total salivary IgA concentrations and the IgA response specific for CarLA. Interestingly, we found an effect of stress on CarLA IgA concentrations but not on total IgA concentrations. Secretory IgA concentrations have been shown to change in response to both physical and psychological stressors in several different species (reviewed by Staley et al. [6]). Therefore, it is unclear why we did not find changes in total IgA concentrations in the present study.

In the present study, an open-field test was used to categorize ewes into two temperament categories, high and low reactors. High reactive ewes showed a greater behavioral (vocalizations and locomotion) and physiological (heart rate) response to the open-field test. In previous studies, vocalizations and locomotion behaviors in response to an arena test have been found to be replicable across time and are therefore considered good behavioral measures that reflect individual differences among animals [33]. In addition, the novel arena/open-field test can be considered a good fear test for sheep [23], and locomotor activity and vocalizations are associated with signs of fear [34]. Therefore, we predicted that the sheep that reacted more in response to an open-field test were more fearful and would find exposure to the visual isolation test more stressful than low reactive sheep. However, we found no significant relationship between stress and temperament in the present study. We instead found that CarLA IgA concentrations were lower in high reactive than low reactive ewes after a 0.5 h testing period, regardless of the testing environment. High reactive ewes may have perceived the testing environment as being more stressful than low reactive animals, whether they were exposed to a visual isolation test or housed in control pens; however, these animals exhibited no corresponding increase in heart rate or cortisol concentrations to support this speculation. Furthermore, Beausoleil et al. [16] found that sheep categorized as being ‘more’ active were actually less fearful and had lower cortisol concentrations in response to the presence of a human in an arena test than ‘less’ active sheep. Alternatively, as several of the ewes would have had the same sire in the present study, there may be a genetic relationship between temperament and antibody activity; sires that produce off-spring with higher antibody activity may also produce offspring that are more fearful. However, it does not appear that the behavioral response of sheep to an arena test is genetically corelated [35] and CarLA IgA responsiveness was shown to be only moderately heritable [13]. In this study we were not able to select ewes that represented the extremes of each temperament category as has been done in other studies [16]; therefore, there may have been a high degree of individual variation among the animals in each of the temperament categories which may have diluted any potential effects that temperament may have had on the variables tested in this study. In future studies, it would be of interest to test animals that represent the extremes of each temperament category and investigate if there is a relationship between stress and temperament or categorize animals based on other temperament traits, such as sociality or boldness.

## 5. Conclusions

Stress appeared to have an immunosuppressive effect on CarLA IgA concentrations in ewes but not on total IgA concentrations. However, there appeared to be no relationship between animal temperament and stress on the humoral immune response. More research is needed to understand the mechanism involved in the immunosuppressive effect of stress on CarLA IgA concentrations and what it means regarding ewe immunity to parasitism, particularly in relation to farm-relevant stressors. Future research should measure antibodies against a range of related and unrelated pathogens (for example, antibodies to Clostridial vaccines, leptospirosis, Salmonella etc.) to provide a picture of stress-induced changes in immunity.

## Figures and Tables

**Figure 1 animals-09-00104-f001:**
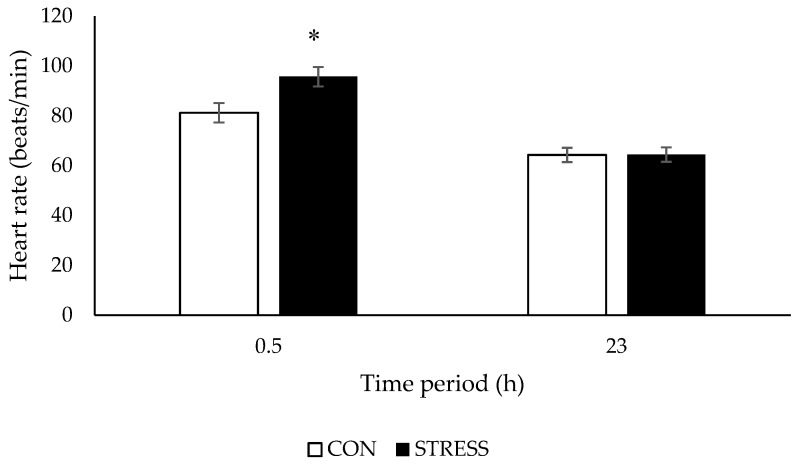
Heart rate (beats/min; means ± SED) of ewes housed in control (CON) or visually isolated pens (STRESS) during the first and last 30 min of a 23 h test period. * For a time period, least square means differ at *p* < 0.05.

**Figure 2 animals-09-00104-f002:**
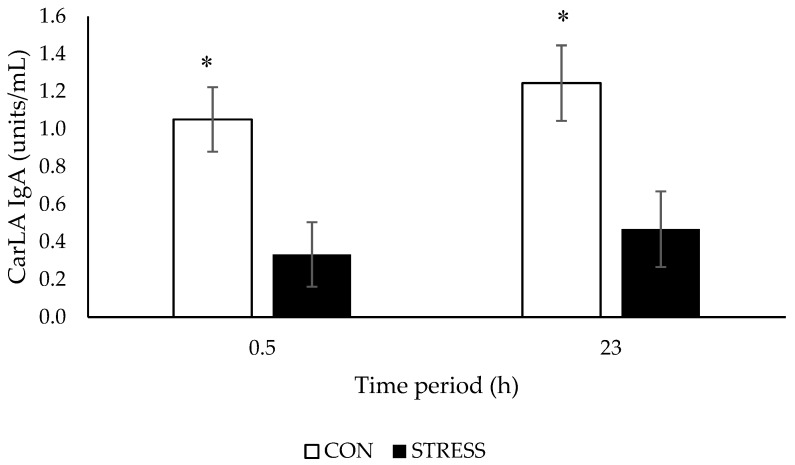
CarLA concentrations (units/mL; means ± SED) of ewes housed in control (CON) or visually isolated pens (STRESS) after a 0.5 h and a 23 h test period. * For a time period, least square means differ at *p* < 0.05 (back-transformed means).

**Figure 3 animals-09-00104-f003:**
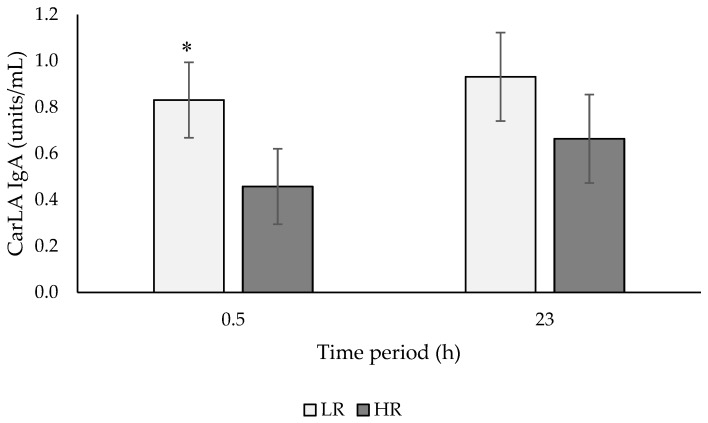
CarLA concentrations (units/mL; means ± SED) of low (LR) and high (HR) ewes after a 0.5 h and a 23 h test period. * For a time period, least square means differ at *p* < 0.05 (back-transformed means).

**Figure 4 animals-09-00104-f004:**
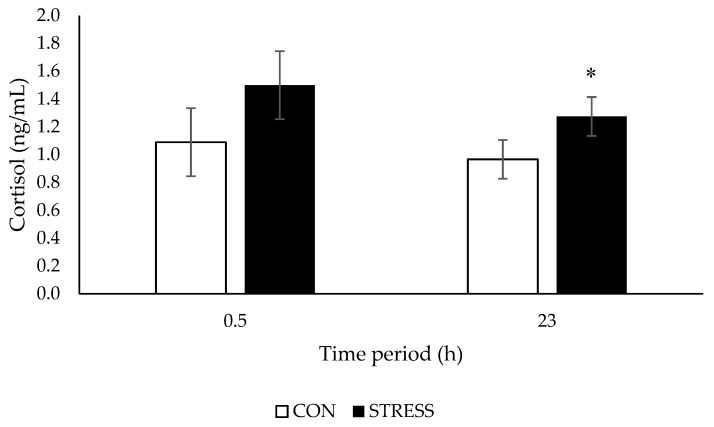
Cortisol concentrations (ng/mL; means ± SED) of ewes housed in control (CON) or visually isolated pens (STRESS) after a 0.5 h and a 23 h test period. * For a time period, least square means differ at *p* < 0.05.

**Table 1 animals-09-00104-t001:** Description of behaviors measured in the open-field and visual isolation tests.

Behavior ^1^	Description
Vocalization	Any vocalization including open- and closed-mouth bleats
Defecation event	Elimination of feces from the body
Urination event	Bending of the back legs and lowering into a squatting position and urinating
Escape attempt	Both front feet are off the ground and placed against the side of the pen
Squares entered	A square was recorded as being entered when both front feet of the ewe had entered the same square
Behavioral score ^2^	Sum of the frequency of vocalizations, urination and defecation events, and escape attempts
Reactivity score ^3^	Sum of the frequency of vocalizations, urination and defecation events, escape attempts and number of squares entered

^1^ All behaviors were measured as frequencies. ^2^ Behavioral score recorded during the first 0.5 h of the visual isolation test. ^3^ Reactivity score used to categorize animals as either low (LR) or high responders (HR) in response to the five-minute open-field test.

**Table 2 animals-09-00104-t002:** Average and range of the heart rate and behavioral response of low (LR) and high (HR) reactive ewes during a five-minute open-field test.

	Sheep Temperament			
Measure	LR		HR	
Change in heart rate (beats/min)	17.5	(−31.3–44.6)	26.3	(−11.7–68.2)
Vocalizations (no./5 min)	7.1	(0–27)	21.1	(0–53)
Urination events (no./5 min)	0.50	(0–2)	0.68	(0–3)
Defecation events (no./5 min)	0.73	(0–2)	0.28	(0–2)
Escape attempts (no./5 min)	0.07	(0–2)	1.52	(0–17)
Squares entered (no./5 min)	28.7	(2–52)	58.7	(29–140)

**Table 3 animals-09-00104-t003:** Behavioral and physiological response of low (LR) and high (HR) reactive ewes housed in control (CON) or visually isolated (STRESS) pens after a 0.5 h and 23 h test period. Eighty ewes were allocated to one of four treatment groups in a 2 × 2 factorial design (*n* = 20 ewes/treatment). CarLA = carbohydrate larval surface antigen.

	Temperament			Stress			*p*-values		
Measure	LR	HR	SED	CON	Stress	SED	Temperament	Stress	Temperament × Stress
0.5 h									
Behavioral score (no./0.5 h) ^1^	10.7	12.4	3.97	0.3	22.9	3.97	0.440	<0.001	0.790
Vocalizations (no./0.5 h) ^1^	10.4	11.9	3.96	0.0	22.3	3.96	0.575	<0.001	0.575
Heart rate (beats/min)	88.3	86.8	3.90	81.2	95.7	3.90	0.740	<0.001	0.399
CarLA IgA (units/mL) ^2^	0.83	0.46	0.163	1.05	0.33	0.172	0.022	<0.001	0.135
Total IgA (units/mL) ^2^	344	273	74.3	296	317	74.3	0.352	0.764	0.673
Cortisol (ng/mL)	1.45	1.14	0.229	1.09	1.50	0.245	0.192	0.098	0.536
23 h									
Heart rate (beats/min)	63.0	66.8	2.87	64.3	64.4	2.87	0.176	0.823	0.145
CarLA IgA (units/mL) ^2^	0.93	0.66	0.191	1.24	0.47	0.201	0.147	<0.001	0.414
Total IgA (units/mL) ^2^	372	290	74.6	351	308	74.6	0.264	0.590	0.729
Cortisol (ng/mL)	1.09	1.15	0.130	0.97	1.28	0.140	0.629	0.031	0.495

^1^ Behavior was only measured during the first 0.5 h of the visual isolation test. ^2^ Back-transformed means.

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
