# Peer review of "Stress-Induced Immunomodulation in Low and High Reactive Sheep"

_animals, 2019, doi:10.3390/ani9030104_

Round 1

Reviewer 1 Report

The paper entitled “Stress-induced immunomodulation in low and high reactive sheep” deals with an interesting topic aimed at studying the linking between stress and immunity (particularly mucosal immunity), and suggest the potentiality of using the Immunoglobulin A as a non-invasive physiological biomarker of stress.

Overall, the paper was well written, however some crucial points have to be dilucidated by authors before its acceptance. The main criticisms are related to the objectives and the discussion of data and to the experimental design.

The specific objective of the manuscript (L98-101) was the study of the relationship between stress and temperament on a selected, specific humoral immune responses of ewes after exposure to a short and longer term stressor. Conversely, at Lines 363-366 the authors affirmed that “in future studies, would be of interest to investigate the relationship between sheep temperament and CarLA IgA concentrations to determine if there is the possibility that significant between-animal variability in such responses could be utilized for selection purposes”. This statement was one of the possible results of the present paper based on the reported objective. It is not clear as a matter of fact which is the main objective of the manuscript. Indeed, in the conclusion section the authors defined that “stress appeared to have an immusuppressive effect on CarLa IgA concentration”, without adding other comments on the other part of the objective related to the relationship between temperament and stress, as reported at line 363-366. In my opinion the authors have to clarify this aspect because of the model studied they can affirm that different behavioral profile can activate different response to stressors event. in addition, in the tables the authors did not present the interaction between temperament (LR and HR) and stress response (CON and STRESS group), even if in the result and discussion sections the statistical results was presented. I suggest to add in the table 3 the results related to CON and STRESS ewes of each LR and HR group of animals and consequently the p-value of the interaction stress x temperament and not only the mean as depicted in the figure 1 (It referred to the mean of LR and HR ewes in the CON group and the mean of LR and HR ewes in the STRESS group). Based on this, the relationship between temperament and stress was already assessed in the experiment and the result was that no relationship was found.

In relation to the experimental design the authors should clarify the methodology of ranking animals based on behavioral responses. This is not clear at all. The authors asserted that the classification of animals was based on a score. Which type of score? How did they decide to classify the animals based on that score? This aspect is crucial for the acceptance of the manuscript.

Minor comment

Table 1- Please add the score related to the classification of animals into LR and HR.

Line 188- Please substitute “substrate” into “TMB” using the extended form.

Line 206- Please add the citation “Shaw et al. 2013) in the numerical form.

Line 216- For the statistical analysis I guess that the authors had to perform the MIXED model of AVOVA considering that the authors explicated both fixed and random effects in the statistical analysis. The authors are invited to fit this point.

Line 292- The authors affirmed that “continued elevation of cortisol concentrations”; however, the cortisol increased in STRESS ewes only after 23h. I think that the authors have to explain better this concept to avoid speculation.

Author Response

The authors would like to thank the reviewer for their thoughtful and very helpful comments on this manuscript. The authors have considered all the comments listed below, have responded to these comments individually and made the appropriate changes to the manuscript, which have been highlighted in yellow.

The paper entitled “Stress-induced immunomodulation in low and high reactive sheep” deals with an interesting topic aimed at studying the linking between stress and immunity (particularly mucosal immunity), and suggest the potentiality of using the Immunoglobulin A as a non-invasive physiological biomarker of stress.

Overall, the paper was well written, however some crucial points have to be dilucidated by authors before its acceptance. The main criticisms are related to the objectives and the discussion of data and to the experimental design.

The specific objective of the manuscript (L98-101) was the study of the relationship between stress and temperament on a selected, specific humoral immune responses of ewes after exposure to a short and longer term stressor. Conversely, at Lines 363-366 the authors affirmed that “in future studies, would be of interest to investigate the relationship between sheep temperament and CarLA IgA concentrations to determine if there is the possibility that significant between-animal variability in such responses could be utilized for selection purposes”. This statement was one of the possible results of the present paper based on the reported objective. It is not clear as a matter of fact which is the main objective of the manuscript. Indeed, in the conclusion section the authors defined that “stress appeared to have an immusuppressive effect on CarLa IgA concentration”, without adding other comments on the other part of the objective related to the relationship between temperament and stress, as reported at line 363-366. In my opinion the authors have to clarify this aspect because of the model studied they can affirm that different behavioral profile can activate different response to stressors event.

Au: The reviewer makes a very good point. In the manuscript we have now focused more on discussing the results in relation to the entire objectives of the study. In the conclusion we now also mention all aspects of the objectives.

in addition, in the tables the authors did not present the interaction between temperament (LR and HR) and stress response (CON and STRESS group), even if in the result and discussion sections the statistical results was presented. I suggest to add in the table 3 the results related to CON and STRESS ewes of each LR and HR group of animals and consequently the p-value of the interaction stress x temperament and not only the mean as depicted in the figure 1 (It referred to the mean of LR and HR ewes in the CON group and the mean of LR and HR ewes in the STRESS group). Based on this, the relationship between temperament and stress was already assessed in the experiment and the result was that no relationship was found.

Au: Table 3 has now being revised and now includes the results for the temperament and stress analysis. Also the p-values for the stress by treatment interaction. The overall treatment means have not being included at there was no stress by treatment interaction, hence only the results for the separate temperament and stress analysis have been presented.

In relation to the experimental design the authors should clarify the methodology of ranking animals based on behavioral responses. This is not clear at all. The authors asserted that the classification of animals was based on a score. Which type of score? How did they decide to classify the animals based on that score? This aspect is crucial for the acceptance of the manuscript.

Au: The methodology used to rank the ewes has now been clarified in the methods section. The method of categorising the animals was based on other published research in sheep investigating the effect of temperament. Furthermore, the definition of the behavioural and reactivity scores has now also been made more clear in Table 1.

Minor comment

Table 1- Please add the score related to the classification of animals into LR and HR.

Au: This information has been added.

Line 188- Please substitute “substrate” into “TMB” using the extended form.

Au: This information has been added.

Line 206- Please add the citation “Shaw et al. 2013) in the numerical form.

Au: Done.

Line 216- For the statistical analysis I guess that the authors had to perform the MIXED model of AVOVA considering that the authors explicated both fixed and random effects in the statistical analysis. The authors are invited to fit this point.

Au: We have deleted the open field test analysis based on reviewer 2 comments, therefore the stats section now reads:

“Data from the OF test are presented descriptively. The data from the visual isolation test were analysed using Genstat 18th edition (VSN International LTD, 2015) as a linear mixed model with day of the test as a random effect, and temperament and stress treatments and their interaction as fixed effects.”

Line 292- The authors affirmed that “continued elevation of cortisol concentrations”; however, the cortisol increased in STRESS ewes only after 23h. I think that the authors have to explain better this concept to avoid speculation.

Au: Cortisol concentration tended to be higher in STRESS ewes at 30 min, so the sentence now reads ‘sheep vocalised more, heart rate was elevated and cortisol concentrations tended to be elevated in response to a 30 min visual isolation test, therefore it appears as though….’

Reviewer 2 Report

This is an original idea from an excellent group. I have some problems with understanding but this could be resolved by rewriting various sections as flagged. One source of confounding that should be discussed is genetic relationships. If several animals share a common ancestor then they could have similar stress responses and  similar antibody activities. Therefore the genetic relationships could create a correlation between antibody activity and stress responses even though there need not be any mechanistic relationship.

Line 26: What does evenly mean? Were equal numbers of the ewes allocated at random to the treatment groups? 

Line 28: salvia → saliva

Lines 53-55: This is not strictly true in ruminants because IgG1 can be the major mucosal antibody is some situations.

Line 57-58: Do you mean enhancing or suppressive?

Line 82: More and less or high and low.

Lines 140-142: This is confusing. Perhaps 4 treatment groups in a 2x2 design or 2 treatment and 2 control groups with 20 animals (HR or LR) in each group.

Lines 216-218: This test is better described as a linear mixed model.

Section 3.1.: It is not appropriate to divide the animals into high and low groups then test for significance between the two groups. You can present the data without the p-values. Better to plot the data as a frequency distribution without attempting statistical analyses.

Section 3.2.: It is not clear why the values in table 3 were summed rather than averaged?

Line 235: This value seems at odds with table 3.

Lines 253-257: This not clear. Were the IgA activities against CarLA measured at time 0? Is there any evidence that there was a drop in IgA activity as a consequence of the stress? Otherwise surely the interpretation is that the animals selected as HR had lower preexisting responses to CarLA?

Lines 301-303: What is the evidence that there was a change in IgA activity?

Lines 303-305: Antibody production takes about 7 days and usually responses refer to this. Do you mean that there was an increase in the concentration of IgA in the saliva that was due to increased transfer rather than increased production?

Lines 338-340: Best to rephrase and say exactly what was found. IgA activity against CarLA was lower in HR ewes compared to LR ewes.

Author Response

The authors would like to thank the reviewer for their thoughtful and very helpful comments on this manuscript. The authors have considered all the comments listed below, have responded to these comments individually and made the appropriate changes to the manuscript, which have been highlighted in yellow.

This is an original idea from an excellent group. I have some problems with understanding but this could be resolved by rewriting various sections as flagged. One source of confounding that should be discussed is genetic relationships. If several animals share a common ancestor then they could have similar stress responses and  similar antibody activities. Therefore the genetic relationships could create a correlation between antibody activity and stress responses even though there need not be any mechanistic relationship.

Au: The reviewer brings up a really good point. It is highly likely that some of the ewes would have had the same sire, therefore there could be a genetic relationship between ewes that are more reactive to an OF test (temperament) and antibody activity.  There was no difference in baseline antibodies between low and high reactive ewes, however, there could still be a link between genetics, temperament and antibody activity. This has been added to the discussion.

Au: This sentence as well as L140 has been revised to make it more clear that this study involved 4 treatments in a 2 x 2 factorial design.

Line 28: salvia → saliva

Au: Corrected

Lines 53-55: This is not strictly true in ruminants because IgG1 can be the major mucosal antibody is some situations.

Au: Thank you for the correction, the sentence now reads ‘Immunoglobulin (Ig) A is one of the major antibody associated with ……….’

Line 57-58: Do you mean enhancing or suppressive?

Au: Yes, or, this has been corrected.

Line 82: More and less or high and low.

Au: Changed to More and Less.

Lines 140-142: This is confusing. Perhaps 4 treatment groups in a 2x2 design or 2 treatment and 2 control groups with 20 animals (HR or LR) in each group.

Au: This sentence as well as L26 has been revised to make it more clear that this study involved 4 treatments in a 2 x 2 factorial design.

Lines 216-218: This test is better described as a linear mixed model.

Au: This section now reads: “Data from the OF test are presented descriptively. The data from the visual isolation test were analysed using Genstat 18th edition (VSN International LTD, 2015) as a linear mixed model with day of the test as a random effect, and temperament and stress treatments and their interaction as fixed effects.”

Section 3.1.: It is not appropriate to divide the animals into high and low groups then test for significance between the two groups. You can present the data without the p-values. Better to plot the data as a frequency distribution without attempting statistical analyses.

Au: The reviewer makes a very good point. The p-values have been removed from the table and text. Instead the average and range of each measure for each temperament has been added to Table 2 to give readers a better indication of the variability amongst categories.

Section 3.2.: It is not clear why the values in table 3 were summed rather than averaged?

Au: Sorry, but I am not completely sure what the reviewer is referring to. In Table 2, ewes were only in the open-field test for 5 mins, so the data in Table 2 represents the mean for each of the behaviours over the entire 5 min testing period. However, in table 3 the behaviour was measured over 30 mins so the data in Table 3 represents the mean for each of the behaviours over the entire 30 min testing period. Table 3 has now been revised to make it more clear to the readier that the study was a 2 x 2 factorial design and as there was no significant stress x temperament interactions, only the main effects of ‘stress’ and ‘ temperament’ are presented. 

Line 235: This value seems at odds with table 3.

Au: This data has been corrected and removed to Table 3. Table 3 has now being revised and now includes the results from the temperament and stress analysis. Also, the p-values for the stress by treatment interaction. The overall treatment means have not being included as there was no stress by treatment interaction, hence only the results for the separate temperament and stress analysis has been presented.

Lines 253-257: This not clear. Were the IgA activities against CarLA measured at time 0? Is there any evidence that there was a drop in IgA activity as a consequence of the stress? Otherwise surely the interpretation is that the animals selected as HR had lower preexisting responses to CarLA?

Au: Yes, saliva samples were collected 5 minutes before sheep entered the treatment pens to obtain baseline values for CarLA, IgA and cortisol concentrations. This information is in the methods section. Baseline values for CarLA, IgA and cortisol were used as covariates for analysis of the 30 min and 23 h post-treatment values to compensate for baseline variation. This information has been added to the stats section. There was no difference in IgA or CarLA level at baseline between LR and HR ewes. IgA levels changed very little over time, however CarLA levels declined in both CON and STRESS sheep, but more so in STRESS sheep. This information has now been added to the results section.

Lines 301-303: What is the evidence that there was a change in IgA activity?

Au: See comment below. This evidence is now included in the results section.

Lines 303-305: Antibody production takes about 7 days and usually responses refer to this. Do you mean that there was an increase in the concentration of IgA in the saliva that was due to increased transfer rather than increased production?

Au: Yes, this has now been clarified and the sentence now reads ‘The short time frame in which an acute stressor can cause changes in S-IgA concentrations is likely due to changes in IgA secretion mediated by the autonomic nervous system’

Lines 338-340: Best to rephrase and say exactly what was found. IgA activity against CarLA was lower in HR ewes compared to LR ewes.

Au: Overall, we did find that stress caused a reduction in CarLA activity, irrespective of ewe temperament. CarLA levels were lower in both CON and STRESS ewes at 30 minutes and 23 h compared to baseline values, however, this reduction was about double in STRESS ewes. This information has now been added to the results section. Table 3 has also been revised and now more clearly shows the ‘stress’ and ‘temperament’ response at 30 minutes and 23 hours. In addition, HR ewes had lower CarLA levels than LR ewes but there was no significant stress by temperament interaction.

Round 2

Reviewer 1 Report

The revised form of the manuscript entitled “Stress-induced immunomodulation in low and high reactive sheep” meets all the reviewer’s suggestions. However, some criticisms emerged in the final part of the Discussion. I strongly suggest to add some references when the authors try to give some explanations on the absence of significant relationship between stress and temperament (L-359-374). The authors should add some sentences such as "our hypothesis was that"…or similar in order to don’t seem speculative and, in particular, should add references when discuss about some specific connections such as at L 366- 368.

Minor comments

L 13: Please delete “an” before “animal’s immune system”.

L 360: Please change “did find” into “found”.

Author Response

The revised form of the manuscript entitled “Stress-induced immunomodulation in low and high reactive sheep” meets all the reviewer’s suggestions. However, some criticisms emerged in the final part of the Discussion. I strongly suggest to add some references when the authors try to give some explanations on the absence of significant relationship between stress and temperament (L-359-374). The authors should add some sentences such as "our hypothesis was that"…or similar in order to don’t seem speculative and, in particular, should add references when discuss about some specific connections such as at L 366- 368.

Au: This paragraph has been expanded upon and now includes a better explanation of the results and more references. These changes have been highlighted in yellow.

Minor comments

L 13: Please delete “an” before “animal’s immune system”.

Au: Done.

L 360: Please change “did find” into “found”.

Au: Changed